# A-Loc: Efficient Alternating Iterative Methods for Locating the $k$ Largest/Smallest Elements in a Factorized Tensor

## Abstract

Tensors, especially higher-order tensors, are typically represented in low-rank formats to preserve the main information of the high-dimensional data while saving memory space. Locating the largest/smallest elements in a tensor with the low-rank format is a fundamental task in a large variety of applications. However, existing algorithms often suffer from low computational efficiency or poor accuracy. In this work, we propose a general continuous optimization model for this task, on top of which an alternating iterative method combined with the maximum block increasing (MBI) approach is presented. Then we develop a novel block-search strategy to further improve the accuracy. The theoretical analysis of the convergence behavior of the alternating iterative algorithm is also provided. Numerical experiments with tensors from synthetic and real-world applications demonstrate that our proposed algorithms achieve significant improvements in both accuracy and efficiency over the existing works.

## 1 Introduction

Large volumes of high-dimensional data, such as simulation data, video, and hyperspectral images, have sprung up in scientific computing, machine learning, and many other applications. These high-dimensional data can be naturally represented by tensors, and are usually stored in the lossy compressed format based on tensor decomposition to alleviate the curse of dimensionality Kolda & Bader (2009); Cong et al. (2015); Sidiropoulos et al. (2017); Cichocki et al. (2016; 2017). In most scenarios, only a few elements of the tensor are of interest. For example, in recommendation systems, the $k$ largest elements correspond to the most meaningful concerns for personalized recommendations Symeonidis (2016); Frolov & Oseledets (2017); Zhang et al. (2021). For quantum simulations, the compressed format of the tensor is usually used to represent the quantum state to reduce memory usage, and we can get the maximum likelihood or maximum a-posteriori estimation to measure the quantum state by locating the largest elements in the compressed tensors. Furthermore, we could solve the top-$k$ elephant network flows problem in the network management field by locating the largest $k$ elements of the low-rank tensor recovered from the observed data Xie et al. (2019). Therefore, it is a crucial problem to find the $k$ largest elements in the compressed format of the tensor in practical applications. Many methods have been proposed to solve this problem Higham & Relton (2016); Grasedyck et al. (2019); Chertkov et al. (2022); Sidiropoulos et al. (2022).

CANDECOMP/PARAFAC (CP) format Hitchcock (1927) is one of the most popular tensor decomposition models for dimensionality reduction and data mining Papalexakis et al. (2016); Tang & Liao (2020). Specifically, given an $N$th-order tensor $\boldsymbol{\mathcal{A}} \in \mathbb{R}^{I_1 \times I_2 \cdots \times I_N}$, its CP decomposition represents it by a sum of $R$ rank-one tensors, namely

$$\boldsymbol{\mathcal{A}} = \sum_{r=1}^{R} \boldsymbol{U}_{:,r}^{(1)} \circ \boldsymbol{U}_{:,r}^{(2)} \cdots \circ \boldsymbol{U}_{:,r}^{(N)}, \tag{1.1}$$

where "$\circ$" represents the vector outer product, $\{\boldsymbol{U}^{(n)} \in \mathbb{R}^{I_n \times R}\}_{n=1}^{N}$ are factor matrices, and $R$ is called the CP-rank of $\boldsymbol{\mathcal{A}}$. In this work, we focus on locating the largest/smallest elements for an $N$th-order tensor $\boldsymbol{\mathcal{A}}$ given in CP format. The proposed algorithms could naturally apply to other tensor formats such as Tucker, tensor-train (TT), and quantized TT (QTT).

**Prior work.** For this task, Lu *et al.* proposed a sampling method, namely star sampling Lu et al. (2017), which is a generalization of the randomized diamond sampling method in the matrix case proposed in Ballard et al. (2015). It is well known that the accuracy of the sampling method strongly depends on the quality of the samples. Since the samples are only determined by factors of the CP format in star sampling, it suffers from low accuracy and instability. A more accurate approach is to convert the task of locating the largest/smallest elements in a factorized tensor to a symmetric eigenvalue problem, which was first proposed by Espig *et al.* Espig et al. (2013). Then the classical power iteration method can be used to find the largest element, see Espig et al. (2013; 2020); Soley et al. (2021). However, it is only suitable for finding the largest element, and cannot directly obtain the corresponding location due to errors in the iterative process. In addition, the Hadamard product in the iterative process will lead to the growth of the CP-rank, which needs to be suppressed by introducing the recompression operation and requires high time and memory costs in practice. Recently, Sidiropoulos *et al.* considered the tensor given in CP format, and proposed an equivalent optimization model Sidiropoulos et al. (2022). The optimization problem is then solved by the proximal gradient descent (PGD) method, but its accuracy highly depends on the choice of hyperparameters such as step size, which is not general in practical applications.

**Contributions.** To solve this problem, inspired by Espig et al. (2013; 2020), we provide a continuous optimization model combined with the rank-one structure of the tensor, which corresponds to the eigenvector. And several algorithms are proposed based on the optimization model. First, we present an alternating iterative method combined with the maximum block increasing (MBI) strategy Chen et al. (2012), and establish its convergence theory. On top of that, we also develop a novel block-search strategy to further improve accuracy. The proposed algorithms have some advantages. On the one hand, our proposed algorithms can obtain the largest/smallest elements and its location simultaneously, and due to the use of rank-one structure, they have a significant improvement in computational efficiency and memory cost compared with power iteration. On the other hand, since the proposed model is more general than the optimization model proposed in Sidiropoulos et al. (2022), our proposed algorithms can be naturally applied to various low-rank representations, such as CP, Tucker, and TT formats, and reduce the dependence on hyperparameters. Numerical experiments demonstrate that our proposed algorithms can achieve significant improvements in both accuracy and efficiency over the existing works.

**Notations.** In this paper, we use boldface lowercase and capital letters (e.g., $\boldsymbol{a}$ and $\boldsymbol{A}$) to denote vectors and matrices. The boldface Euler script letter is used to denote higher-order tensors, e.g., an $N$th-order tensor can be expressed as $\boldsymbol{\mathcal{A}} \in \mathbb{R}^{I_1 \times I_2 \cdots \times I_N}$, where $I_n$ denotes the dimension of mode-$n$, and the $(i_1, i_2, \cdots, i_N)$-th element of it is represented by $\boldsymbol{\mathcal{A}}_{i_1, i_2, \cdots, i_N}$.

## 2 PERSPECTIVE OF CONTINUOUS OPTIMIZATION MODEL

As already mentioned in Espig et al. (2013), locating the largest/smallest elements of $\boldsymbol{\mathcal{A}}$ is equivalent to solving the corresponding symmetric eigenvalue problem. Further, we propose a general continuous optimization model based on the rank-one structure of the tensor corresponding to the eigenvector. Let $\boldsymbol{\mathcal{A}} \in \mathbb{R}^{I_1 \times I_2 \cdots \times I_N}$ be an $N$th-order tensor in the CP format 1.1. Without loss of generality, we assume that the tensor $\boldsymbol{\mathcal{A}}$ is non-negative. If not, non-negativity can be satisfied by shift transformation, that is, $\boldsymbol{\mathcal{A}} + s\boldsymbol{\mathcal{E}}$, where $s > 0$ is large enough such as $s = \|\boldsymbol{\mathcal{A}}\|_F$, and $\boldsymbol{\mathcal{E}}$ is a tensor whose elements are all 1. Due to the rank-one structure of $\boldsymbol{\mathcal{E}}$, the CP-rank of the transformed tensor $\boldsymbol{\mathcal{A}} + s\boldsymbol{\mathcal{E}}$ is at most $R+1$. Our work is to find the largest and smallest elements in $\boldsymbol{\mathcal{A}}$, along with their locations. Take locating the largest element as an example, and a similar result can be derived for the smallest element. We first define a diagonal matrix $\boldsymbol{A} \in \mathbb{R}^{I_1 I_2 \cdots I_N \times I_1 I_2 \cdots I_N}$ such that

$$\boldsymbol{A}_{i,i} = \boldsymbol{\mathcal{A}}_{i_1, i_2, \cdots, i_N} \text{ with } i = i_1 + \sum_{j=1}^{N-1} (i_{j+1} - 1) I_{1:j},$$

where $I_{1:j} = \prod_{k=1}^{j} I_k$. Then the largest eigenpair of $\boldsymbol{A}$ corresponds to the largest element of $\boldsymbol{\mathcal{A}}$ and its location, which can be obtained by solving the following spherical constrained optimization problem

$$\max_{\boldsymbol{x} \in \mathbb{R}^{I_1 I_2 \cdots I_N}} \boldsymbol{x}^T \boldsymbol{A} \boldsymbol{x} \text{ subject to } \|\boldsymbol{x}\|_2 = 1. \tag{2.1}$$

In tensor notations, problem 2.1 can be rewritten as

$$\max_{\boldsymbol{\mathcal{X}} \in \mathbb{R}^{I_1 \times I_2 \cdots \times I_N}} \langle \boldsymbol{\mathcal{X}}, \boldsymbol{\mathcal{A}} * \boldsymbol{\mathcal{X}} \rangle \text{ subject to } \|\boldsymbol{\mathcal{X}}\|_F = 1, \tag{2.2}$$

where "$*$" represents the Hadamard product of tensors. Due to the diagonal structure of the symmetric matrix $\boldsymbol{A}$, we know that the $N$th-order tensor $\mathcal{X}$ corresponding to the solution of problem 2.2 has a rank-one structure. Therefore, combined with the CP representation of $\mathcal{A}$, problem 2.2 can be further simplified, see Theorem 1.

**Theorem 1.** *Let $\mathcal{A}$ be an $N$th-order tensor, and its CP format be described in 1.1, then locating the largest element of $\mathcal{A}$ is equivalent to solving the following spherical constrained optimization problem*

$$\max_{\boldsymbol{x}^{(1)}, \cdots, \boldsymbol{x}^{(N)}} \sum_{r=1}^{R} (\boldsymbol{x}^{(1)T}(\boldsymbol{U}_{:,r}^{(1)} * \boldsymbol{x}^{(1)})) \cdots (\boldsymbol{x}^{(N)T}(\boldsymbol{U}_{:,r}^{(N)} * \boldsymbol{x}^{(N)})) \tag{2.3}$$

$$\text{s.t. } \|\boldsymbol{x}^{(n)}\|_2 = 1 \text{ for all } n = 1, \cdots, N.$$

The proof of Theorem 1 can be found in the Appendix.

From Theorem 1, we know that locating the largest element of the factorized tensor $\mathcal{A}$ can be converted to a continuous optimization problem with spherical constraints, and the calculation of the objective function in problem 2.3 only involves vector-vector multiplications of size $I_n$ for all $n = 1, 2, \cdots, N$, which opens the door to developing algorithms from an optimization perspective. It is clear that the largest element and its location can be directly obtained by the optimal solution of 2.3. In addition, if the smallest element is required, we just replace the maximization in problem 2.3 with the minimization. We remark that Theorem 1 also applies to other tensor formats such as Tucker, TT, and QTT, whose corresponding continuous optimization models can be similarly given. We leave them for future work.

## 3 COMPONENTS OF A-LOC

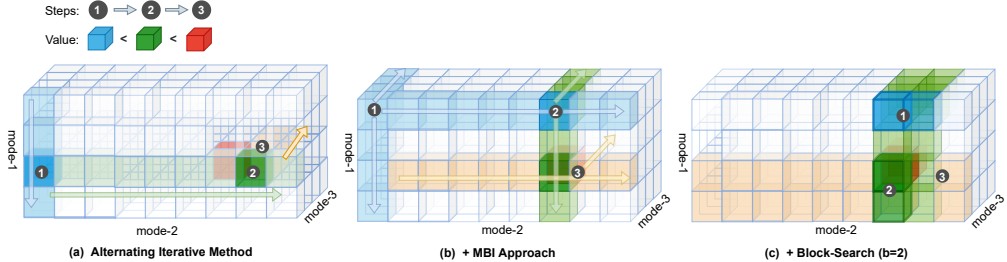

Figure 1: Illustration of the iterative process of A-Loc for locating the largest element in a third-order tensor. Subfigures (a), (b), and (c) depict the steps of the plain alternating iterative algorithm, the incorporation of the MBI approach (named **+MBI Approach**), as well as the addition of a block-search strategy (named **+Block-Search**), respectively. The dark cube represents the maximum value on the current mode and the red cube represents the target value.

A straightforward approach to problem 2.3 is the alternating iterative method, which is parameter-free, easy to implement, and high performance. During the iterations of the alternating iterative method, only one variable is updated at once instead of all of them. For example, all variables are fixed except $\boldsymbol{x}^{(n)}$, then its corresponding subproblem is as follows

$$\max_{\boldsymbol{x}^{(n)}} \sum_{r=1}^{R} \alpha_r \boldsymbol{x}^{(n)T}(\boldsymbol{U}_{:,r}^{(n)} * \boldsymbol{x}^{(n)}) \text{ subject to } \|\boldsymbol{x}^{(n)}\|_2 = 1, \tag{3.1}$$

where $\alpha_r = \prod_{m \neq n} \boldsymbol{x}^{(m)T}(\boldsymbol{U}_{:,r}^{(m)} * \boldsymbol{x}^{(m)})$. Obviously, solving subproblem 3.1 is equivalent to locating the largest element of the vector $\boldsymbol{y} = \sum_{r=1}^{R} \alpha_r \boldsymbol{U}_{:,r}^{(n)} \in \mathbb{R}^{I_n}$, only requires $\mathcal{O}(RI_n)$ time cost.

To further enhance the convergence and accuracy of the plain alternating iterative algorithm, we proposed the incorporation of two techniques: the MBI (Maximal Block Improvement) approach and the block-search strategy. Fig. 1 outlines the iterative process that involves these algorithms

using a third-order tensor as an example. As shown in Fig. 1(a), when the predetermined order $\boldsymbol{x}^{(1)} \to \boldsymbol{x}^{(2)} \to \boldsymbol{x}^{(3)}$ is used to update variables in the plain alternating iterative algorithm, the largest element (i.e., the red cube) may not be detected. In contrast, the MBI approach updates the variable corresponding to the current optimal mode, as illustrated in Fig. 1(b), resulting in the detection of the largest element. Furthermore, as depicted in Fig. 1(c), the block-search strategy stores the $b$ largest elements along the optimal mode, enabling it to detect the largest element. In Sec. 3.3, we will also demonstrate how these proposed algorithms can be extended to locate $k$ largest elements when $k > 1$.

## 3.1 MBI Approach for Convergence

In the one-loop iteration of the plain alternating iterative method, there are $N!$ selections for the updated order of variables $\{\boldsymbol{x}^{(n)}\}_{n=1}^{N}$, and how to select an appropriate order for this method is a crucial issue in practice. As referred in Chen et al. (2012), if the updated order is predetermined and then we adhere to it, the alternating iterative method may fail to converge at all. To address this issue, we introduce an ordering-free approach proposed by Chen et al. (2012), namely MBI, to ensure the convergence of the alternating iterative method. The main idea is to select the variable to be updated so that its corresponding eigenvalue is the largest. Algorithm 1 describes the detailed procedure of the alternating iterative method with the MBI approach.

---

**Algorithm 1** Alternating iterative method with the MBI approach.

---

**Input:** Tensor given in the CP format $\boldsymbol{\mathcal{A}} = \sum\limits_{r=1}^{R} \boldsymbol{U}_{:,r}^{(1)} \circ \boldsymbol{U}_{:,r}^{(2)} \cdots \circ \boldsymbol{U}_{:,r}^{(N)}$, and initial vectors $\{\boldsymbol{x}^{(n)}\}_{n=1}^{N}$

    satisfy $\|\boldsymbol{x}^{(n)}\|_2 = 1$ for all $n = 1, 2, \cdots, N$.

**Output:** The largest element of $\boldsymbol{\mathcal{A}}$ and its corresponding location: $a$ and $\{\boldsymbol{x}^{(n)}\}_{n=1}^{N}$.

1: **while** not convergent **do**
2:    $\boldsymbol{a}, \boldsymbol{i} \leftarrow []$
3:    **for** $n = 1$ to $N$ **do**
4:       $\boldsymbol{q} \leftarrow \sum\limits_{r=1}^{R} \alpha_r \boldsymbol{U}_{:,r}^{(n)}$ where $\alpha_r = \prod\limits_{m \neq n} (\boldsymbol{x}^{(m)T}(\boldsymbol{U}_{:,r}^{(m)} * \boldsymbol{x}^{(m)}))$
5:       $a, i_a \leftarrow$ the largest element in $\boldsymbol{q}$ and its corresponding location
6:       $\boldsymbol{a} \leftarrow [\boldsymbol{a}, a], \boldsymbol{i} \leftarrow [\boldsymbol{i}, i_a]$
7:    **end for**
8:    Find the largest element in $\boldsymbol{a}$ and its corresponding location in $\boldsymbol{i}$: $a$ and $i_a$
9:    $\boldsymbol{x}^{(n_*)} \leftarrow \boldsymbol{e}_{i_a}$ where $n_*$ corresponds to the location of $a$ in $\boldsymbol{a}$, and $\boldsymbol{e}_{i_a}$ is the $i_a$-th column of $I_n \times I_n$ identity matrix
10: **end while**

---

In each loop, the determination of the optimal variable to be updated only depends on factor vectors of the previous iteration step, which is equivalent to the one-loop of Jacobi iteration whose complexity is $\mathcal{O}(\sum\limits_{n=1}^{N} I_n R)$. Due to the dependency-free technique in Jacobi iteration, the *for* loop in Algorithm 1 can be further accelerated in parallel.

In addition, the iterative procedure of Algorithm 1 is essentially equivalent to alternatively searching for the largest element from the fibers of $\boldsymbol{\mathcal{A}}$ along different modes, which is also equivalent to the power iteration proposed in Higham & Relton (2016) for the case of second-order tensors, i.e., matrices. A counterexample is given in Higham & Relton (2016) illustrates that Algorithm 1 may fall into local optima so that the largest element cannot be found, which is the main reason for the low accuracy of the method in some cases.

## 3.2 Block-Search Strategy for Better Accuracy

Further, we develop a novel block-search strategy to improve the accuracy of Algorithm 1, which searches for the largest element from multiple fibers of $\boldsymbol{\mathcal{A}}$ instead of one during the iterations. The detailed computational procedure is described in Algorithm 2.

The first iteration of Algorithm 2 is the same as Algorithm 1, the difference is that we retain the locations corresponding to the $b$ largest elements in $\boldsymbol{q}$, i.e., line 4 of Algorithm 2. Then we search for

---

**Algorithm 2** Algorithm 1 with the block-search strategy.

---

**Input:** Tensor given in the CP format $\mathcal{A} = \sum\limits_{r=1}^{R} \boldsymbol{U}_{:,r}^{(1)} \circ \boldsymbol{U}_{:,r}^{(2)} \cdots \circ \boldsymbol{U}_{:,r}^{(N)}$, block size $b$, and initial

factors $\{\boldsymbol{x}^{(n)}\}_{n=1}^{N}$ satisfy $\|\boldsymbol{x}^{(n)}\|_2 = 1$ for all $n = 1, 2, \cdots, N$.

**Output:** The largest element of $\mathcal{A}$ and its corresponding location: $a$ and $\{\boldsymbol{x}^{(n)}\}_{n=1}^{N}$.

1: $\boldsymbol{x}^{(n_*)} \leftarrow$ the optimal update variable obtained by the MBI strategy

2: $\boldsymbol{q} \leftarrow \sum\limits_{r=1}^{R} \alpha_r \boldsymbol{U}_{:,r}^{(n_*)}$ where $\alpha_r = \prod\limits_{m \neq n_*} \boldsymbol{x}^{(m)T}(\boldsymbol{U}_{:,r}^{(m)} * \boldsymbol{x}^{(m)})$

3: $a \leftarrow$ the largest element in $\boldsymbol{q}$

4: $\boldsymbol{i} \leftarrow$ locations corresponding to the $b$ largest elements in $\boldsymbol{q}$

5: $\boldsymbol{X}^{(n_*)} \leftarrow [\boldsymbol{e}_{\boldsymbol{i}_1}, \cdots, \boldsymbol{e}_{\boldsymbol{i}_b}]$ where $\boldsymbol{e}_{\boldsymbol{i}_j}$ is the $\boldsymbol{i}_j$-th column of $I_{n_*} \times I_{n_*}$ identity matrix

6: **while** not convergent **do**

7:    **for** $n = 1$ to $N$ **do**

8:       **if** $n \neq n_*$ **then**

9:          **for** $j = 1$ to $b$ **do**

10:             $\boldsymbol{q}^j \leftarrow \sum\limits_{r=1}^{R} \alpha_r^j \boldsymbol{U}_{:,r}^{(n)}$ where $\alpha_r^j = (\boldsymbol{X}_{:,j}^{(n_*)T}(\boldsymbol{U}_{:,r}^{(n_*)} * \boldsymbol{X}_{:,j}^{(n_*)})) \prod\limits_{m \neq n_*, n} \boldsymbol{x}^{(m)T}(\boldsymbol{U}_{:,r}^{(m)} *$

            $\boldsymbol{x}^{(m)})$

11:             $a^j \leftarrow$ the largest elements in $\boldsymbol{q}^j$

12:          **end for**

13:          $a, j \leftarrow$ the largest element in $\{a^j\}_{j=1}^{b}$ and its corresponding location in $\{j\}_{j=1}^{b}$

14:          $\boldsymbol{a} \leftarrow [\boldsymbol{a}, a], \boldsymbol{j} \leftarrow [\boldsymbol{j}, j]$

15:       **end if**

16:    **end for**

17:    $m_*, j_* \leftarrow$ the mode and index corresponding to the largest element in $\boldsymbol{a}$

18:    $\boldsymbol{x}^{(n_*)} \leftarrow \boldsymbol{X}_{:,j_*}^{(n_*)}$

19:    $n_* \leftarrow m_*$

20:    $\boldsymbol{q} \leftarrow \sum\limits_{r=1}^{R} \alpha_r \boldsymbol{U}_{:,r}^{(n_*)}$ where $\alpha_r = \prod\limits_{m \neq n_*} \boldsymbol{x}^{(m)T}(\boldsymbol{U}_{:,r}^{(m)} * \boldsymbol{x}^{(m)})$

21:    $a \leftarrow$ the largest element in $\boldsymbol{q}$

22:    $\boldsymbol{i} \leftarrow$ locations corresponding to the $b$ largest elements in $\boldsymbol{q}$

23:    $\boldsymbol{X}^{(n_*)} \leftarrow [\boldsymbol{e}_{\boldsymbol{i}_1}, \cdots, \boldsymbol{e}_{\boldsymbol{i}_b}]$ where $\boldsymbol{e}_{\boldsymbol{i}_j}$ is the $\boldsymbol{i}_j$-th column of $I_{n_*} \times I_{n_*}$ identity matrix

24: **end while**

---

the largest element from $(N-1)b$ fibers of $\mathcal{A}$ in the second iteration, i.e., lines 7-17 of Algorithm 2. Once it is determined, the variable to be updated is determined accordingly, and we also retain the $b$ largest elements. Simultaneously, in order to avoid the explosion of the search space during the iterations, only the column of $\boldsymbol{X}^{(n_*)}$ corresponding to the largest element is reserved as the $n_*$-th factor vector $\boldsymbol{x}^{(n_*)}$, i.e., line 18 of Algorithm 2, which ensures that in each iteration the largest element is found from $(N-1)b$ fibers of $\mathcal{A}$. Compared with Algorithm 1, Algorithm 2 searches for the largest element from a larger space, which is intuitively more likely to obtain the largest element, thereby improving accuracy. It is worth mentioning that although the time cost of Algorithm 2 increases from $\mathcal{O}(\sum\limits_{n=1}^{N} I_n R)$ to $\mathcal{O}(\sum\limits_{n=1}^{N} I_n R b)$ in one-loop iteration, which grows linearly with $N$, it could still maintain high performance as shown in the Experiments.

## 3.3 FINDING THE $k$ LARGEST ELEMENTS

In many scenarios, the $k$ largest elements of the factorized tensor $\mathcal{A}$ are required, thus we present a greedy strategy to find the $k$ largest elements. Suppose that the largest element of $\mathcal{A}$ and its location are obtained and denoted as $a$ and $\boldsymbol{i}$, the optimization problem corresponding to locating the second

largest element can be written as

$$\max_{\boldsymbol{x}^{(1)},\cdots,\boldsymbol{x}^{(N)}} \sum_{r=1}^{R} (\boldsymbol{x}^{(1)T}(\boldsymbol{U}_{:,r}^{(1)} * \boldsymbol{x}^{(1)})) \cdots (\boldsymbol{x}^{(N)T}(\boldsymbol{U}_{:,r}^{(N)} * \boldsymbol{x}^{(N)}))$$

$$\text{s.t. } \|\boldsymbol{x}^{(n)}\|_2 = 1 \text{ for all } n = 1, \cdots, N, \tag{3.2}$$

$$\prod_{n=1}^{N} \langle \boldsymbol{x}^{(n)}, \boldsymbol{e}_{\boldsymbol{i}_n} \rangle = 0,$$

where $\boldsymbol{e}_{\boldsymbol{i}_n}$ represents the $\boldsymbol{i}_n$-th column of the $I_n \times I_n$ identity matrix. Compared to problem 2.3, there is one more constraint in problem 3.2, which increases the difficulty of solving the subproblem that appears in the alternating iterative algorithm. To this end, we apply a shift transformation to the tensor $\boldsymbol{\mathcal{A}}$, i.e., $\boldsymbol{\mathcal{A}} - a\boldsymbol{e}_{\boldsymbol{i}_1} \circ \boldsymbol{e}_{\boldsymbol{i}_2} \cdots \circ \boldsymbol{e}_{\boldsymbol{i}_N}$, so that its largest element corresponds to the second largest element of $\boldsymbol{\mathcal{A}}$, which can then be found by the proposed algorithms. Similarly, other elements can also be sequentially obtained in this way. It is worth mentioning that the CP-rank will increase when the shift transformation is performed, thus the proposed greedy strategy is only suitable for small $k$. For the case of large $k$, it still remains a challenge to develop efficient algorithms.

## 4 CONVERGENCE ANALYSIS

A notable characteristic of alternating iterative algorithms is that the sequence of eigenvalues obtained exhibits a monotonically increasing property, but it does not guarantee convergence to the value corresponding to the stationary point when the update order of variables $\{\boldsymbol{x}^{(n)}\}_{n=1}^{N}$ is predetermined. Thanks to the MBI approach, it guarantees convergence of the alternating iterative algorithm by selecting the optimal variable for updating. The global convergence of Algorithm 1 is illustrated by Theorem 2.

**Theorem 2.** *Let $\{(\boldsymbol{x}_k^{(1)}, \boldsymbol{x}_k^{(2)}, \cdots, \boldsymbol{x}_k^{(N)})\}$ be the sequence obtained by Algorithm 1 with a given initial guess $(\boldsymbol{x}_0^{(1)}, \boldsymbol{x}_0^{(2)}, \cdots, \boldsymbol{x}_0^{(N)})$, and its corresponding sequence of eigenvalues be $\{\lambda_k = \sum_{r=1}^{R} (\boldsymbol{x}_k^{(1)T}(\boldsymbol{U}_{:,r}^{(1)} * \boldsymbol{x}_k^{(1)})) \cdots (\boldsymbol{x}_k^{(N)T}(\boldsymbol{U}_{:,r}^{(N)} * \boldsymbol{x}_k^{(N)}))\}$. Then the sequence $\{\lambda_k\}$ converges to $\lambda_*$ that corresponds to a stationary point of the optimization problem 2.3.*

From Theorem 2, we know that Algorithm 1 converges to the stationary point of the optimization problem 2.3 for any initial guess, but the stationary point is not necessarily optimal. To this end, we further provide a local convergence theory for Algorithm 1, which illustrates that Algorithm 1 converges to the optimal solution with a linear convergence rate when the initial guess is good enough, see Theorem 3.

**Theorem 3.** *Let $(\boldsymbol{x}_*^{(1)}, \boldsymbol{x}_*^{(2)}, \cdots, \boldsymbol{x}_*^{(N)})$ be the unique optimal solution of the optimization problem 2.3, and $\lambda_*$ be the corresponding eigenvalue. If the initial guess $(\boldsymbol{x}_0^{(1)}, \boldsymbol{x}_0^{(2)}, \cdots, \boldsymbol{x}_0^{(N)})$ is sufficiently close to $(\boldsymbol{x}_*^{(1)}, \boldsymbol{x}_*^{(2)}, \cdots, \boldsymbol{x}_*^{(N)})$, then the sequence $\{(\boldsymbol{x}_k^{(1)}, \boldsymbol{x}_k^{(2)}, \cdots, \boldsymbol{x}_k^{(N)})\}$ obtained by Algorithm 1 is R-linearly convergent to $(\boldsymbol{x}_*^{(1)}, \boldsymbol{x}_*^{(2)}, \cdots, \boldsymbol{x}_*^{(N)})$.*

The proofs of Theorem 2 and 3 are provided fully in the Appendix.

## 5 NUMERICAL EXPERIMENTS

To demonstrate the effectiveness of the proposed algorithms, a series of numerical experiments using both synthetic and real-world tensors are carried out in this section. The algorithms we proposed are compared against several baseline methods, including power iteration Espig et al. (2013; 2020), star sampling Lu et al. (2017), and MinCPD via Frank-Wolfe Sidiropoulos et al. (2022). During the power iteration, if the CP-rank of the tensor corresponding to the eigenvector exceeds 10, a recompression operation is performed to suppress rank growth. For the star sampling method, the number of nodes and samples are set to 2 and $\min(10^5, \lfloor 20\% \times \#P(\boldsymbol{\mathcal{A}}) \rfloor)$ respectively, following the guidelines in Lu et al. (2017), where $\#P(\boldsymbol{\mathcal{A}})$ represents the total number of parameters in tensor $\boldsymbol{\mathcal{A}}$. For the MinCPD method, the curvature parameter is set to 5, following the configuration used in Sidiropoulos et al.

(2022). All tested iterative algorithms are terminated when the difference between the eigenvalues from two consecutive iterations falls below a tolerance of $10^{-12}$ or the number of iterations exceeds 1000. Additionally, to minimize the impact of initialization, a restart strategy is utilized in MinCPD and our proposed methods, where the number of restarts is set to 100. For each restart, the initial value is set as the previous result with a random perturbation added. All experiments are conducted on a laptop with an Intel Core i7-11390H CPU (3.40 GHz) and 16GB of memory, and the tensor operations required for the implemented algorithms are coded using the *TensorLy* package, which utilizes NumPy as the backend Kossaifi et al. (2016).

## 5.1 Tests on Random Tensors

In the first example, the accuracy of our proposed algorithms is compared to star sampling and MinCPD methods on randomly generated tensors. Accuracy is defined as $\frac{\#\text{hit}}{S}$, where #hit is the number of times the largest/smallest elements are found, and $S = 50$ is the number of random tensors. The factor matrices $\{U^{(n)} \in \mathbb{R}^{I_n \times R}\}_{n=1}^N$ of each tensor are randomly generated following a Gaussian distribution. The dimensions $I_n$ $(n = 1, 2, \cdots, N)$ and rank $R$ are randomly chosen integers between $[10, 50]$ and $[1, 10]$ respectively.

Table 1 and Fig. 2 depict the accuracy and running time of the tested algorithms. As can be seen from Table 1, the accuracy of the proposed algorithms is significantly higher than MinCPD, especially when using the block-search strategy. Specifically, accuracy improvements of up to $48.2\%$ and $266.7\%$ are achieved in locating the largest and smallest elements, respectively. The results in this table also demonstrate that Algorithm 2 is insensitive to block size $b$, as similar accuracy improvements are achieved with different values of $b$ $(b = 3, 5, 7)$. Besides, while star sampling achieves higher accuracy than the iterative methods in this example, Fig. 2 shows it requires $2.2\times \sim 86.4\times$ more time costs than the proposed algorithms. Another limitation of star sampling is that it is not suitable for locating the smallest element.

Table 1: The obtained accuracy on random tensors for all tested algorithms.

| Algorithms | | $N = 3$ | $N = 4$ | $N = 5$ |
|---|---|---|---|---|
| | Star sampling | **1.00** | **1.00** | **0.96** |
| | MinCPD | 0.54 | 0.54 | 0.56 |
| Maximum | Our | 0.62 | 0.64 | 0.60 |
| | Our ($b = 3$) | 0.76 | 0.70 | 0.66 |
| | Our ($b = 5$) | 0.80 | 0.68 | 0.72 |
| | Our ($b = 7$) | 0.78 | 0.72 | 0.74 |
| | MinCPD | 0.50 | 0.54 | 0.18 |
| | Our | 0.50 | 0.50 | 0.22 |
| Minimum | Our ($b = 3$) | **0.90** | 0.86 | **0.66** |
| | Our ($b = 5$) | **0.90** | **0.90** | 0.62 |
| | Our ($b = 7$) | 0.84 | 0.82 | 0.46 |

## 5.2 Tests on Tensors from Multivariate Functions

In the second example, we examine the efficiency of the proposed algorithms by several large-scale tensors constructed by two multivariate functions, i.e., Rastrigin and Schwefel functions [1]

$$\text{Rastrigin function: } f(\boldsymbol{x}) = 10d + \sum_{i=1}^{d}(x_i^2 - 10\cos(2\pi x_i)), \ x_i \in [-5.12, 5.12],$$

$$\text{Schwefel function: } f(\boldsymbol{x}) = 418.9829d - \sum_{i=1}^{d} x_i \sin(\sqrt{|x_i|}), \ x_i \in [-500, 500],$$

(5.1)

where $d$ is the dimension. To make the size of tensors large enough, we set the dimension $d$ and grid size to 10 and 4096, i.e., the input tensor $\mathcal{A} \in \mathbb{R}^{4096 \times 4096 \cdots \times 4096}$, its CP representation can be derived from Eq. 5.1.

Table 2 records the tested results, including the obtained maximum/minimum value, the number of iterations, and the running time for the tested algorithms. Because power iteration and star sampling

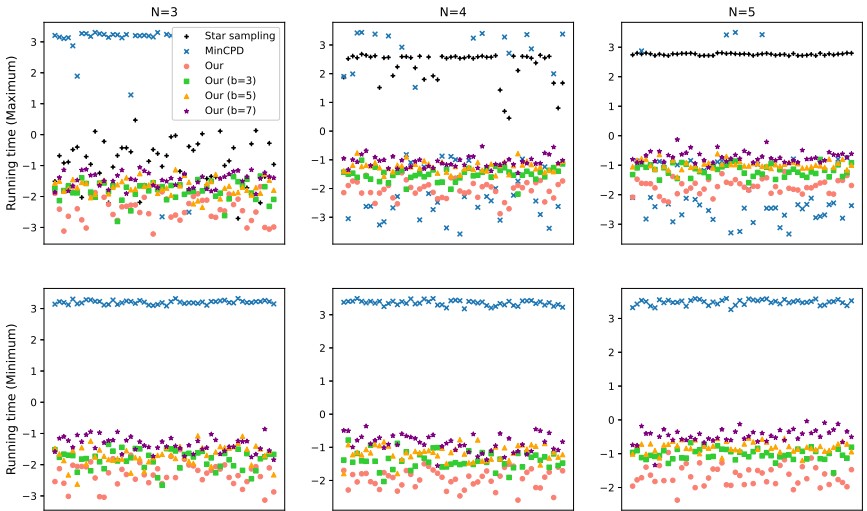

Figure 2: Running time (s) on random tensors for all tested algorithms.

are only suitable for locating the maximum value, we leave the results of minimum as '−'. From the accuracy perspective, it is seen that the proposed algorithms can obtain similar or even better results than their counterparts. Additionally, it is worth mentioning that the accuracy of star sampling is broken due to the increase in the size of tensors. As for performance, our proposed algorithms not only have a faster convergence speed but also are more efficient across all tested algorithms. Table 4 shows that the proposed algorithms can achieve $41.9\times \sim 176.0\times$, $7.4\times \sim 27.7\times$, and $11.02\times \sim 778.52\times$ speedups over the power iteration, star sampling, and MinCPD, respectively.

Table 2: The obtained maximum/minimum value, the number of iterations, and the running time (s) on tensors from multivariate functions for all tested algorithms.

|  | Algorithms | Max / Min | #Iterations | Time |
|---|---|---|---|---|
| Rastrigin | Power iteration | 403.53 / − | 1000 / − | 460.02 / − |
|  | Star sampling | 320.48 / − | − / − | 81.62 / − |
|  | MinCPD | 403.36 / 0.0031 | 1000 / 1000 | 2351.12 / 2291.15 |
|  | Our | **403.53 / 0.0031** | **11 / 11** | **3.02 / 3.02** |
|  | Our ($b = 5$) | 403.53 / 0.0031 | 11 / 11 | 10.99 / 10.92 |
| Schwefel | Power iteration | 8359.64 / − | 1000 / − | 498.21 / − |
|  | Star sampling | 6953.63 / − | 1000 / − | 78.49 / − |
|  | MinCPD | 8375.27 / 0.0103 | 13 / 1000 | 31.20 / 2315.15 |
|  | Our | **8379.65 / 0.0103** | **11 / 11** | **2.83 / 3.07** |
|  | Our ($b = 5$) | 8379.65 / 0.0103 | 11 / 11 | 10.54 / 11.10 |

## 5.3 Tests on Real-World Tensors

In the third example, the accuracy of the proposed algorithms for locating the $k$ largest elements is evaluated on four real-world tensors from various applications, as summarized in Table 3. The tensor 'Cavity' is generated from computational fluid dynamics simulations of lid-driven cavity flow, a standard benchmark problem Burggraf (1966). 'Boltzmann' represents a set of distribution data from Boelens et al. (2020). CP representations of these tensors are obtained using the alternating

---

[1]Retrieved from `http://www-optima.amp.i.kyoto-u.ac.jp/member/student/hedar/Hedar_files/TestGO.htm`

least squares (CP-ALS) method in *TensorLy*. And the accuracy is defined as $\frac{\#\text{hit}}{k}$, where $\#$hit is the number of values found by each algorithm that are smaller than the $k$-th largest element.

The accuracy and running time of the tested algorithms are presented in Table 4. From this table, we observe that Algorithm 2 maintains high accuracy in locating the 5, 10, and 15 largest elements across different real-world tensors. It improves accuracy by at least $14.3\%$ and $25\%$ compared to star sampling and MinCPD, respectively. Additionally, the algorithms we proposed demonstrate substantially lower running times, indicating their suitability for low-latency scenarios.

Table 3: Summary of real-world tensors from various applications.

| Name | Order | CP-rank | Dimension |
|---|---|---|---|
| COVID[2] | 3 | 20 | $438 \times 6 \times 11$ |
| Mnist[3] | 4 | 5 | $28 \times 28 \times 5000 \times 10$ |
| Cavity | 3 | 10 | $512 \times 512 \times 100$ |
| Boltzmann | 4 | 10 | $64 \times 64 \times 64 \times 64$ |

Table 4: The obtained accuracy (%) and running time (s) for locating the 5, 10, and 15 largest elements on real-world tensors for all tested algorithms.

| Algorithms | | top-5 | | top-10 | | top-15 | |
|---|---|---|---|---|---|---|---|
| | | Acc. | Time | Acc. | Time | Acc. | Time |
| COVID | Star sampling | 80.0 | 4.13 | 50.0 | 8.56 | 46.7 | 13.17 |
| | MinCPD | 60.0 | 330.76 | 70.0 | 670.94 | 66.7 | 1060.43 |
| | Our | 20.0 | **1.09** | 40.0 | **2.36** | 53.3 | **3.70** |
| | Our ($b = 5$) | **80.0** | 2.04 | **80.0** | 4.34 | **93.3** | 6.44 |
| Mnist | Star sampling | 0.0 | 93.69 | 10.0 | 194.12 | 6.7 | 309.26 |
| | MinCPD | 0.0 | 1407.67 | 0.0 | 3024.72 | 0.0 | 4920.46 |
| | Our | 0.0 | 1.48 | 30.0 | **3.31** | 53.3 | **5.76** |
| | Our ($b = 5$) | **40.0** | 3.58 | **70.0** | 8.07 | **73.3** | 14.17 |
| Cavity | Star sampling | 0.0 | 63.84 | 0.0 | 131.09 | 13.3 | 204.89 |
| | MinCPD | 20.0 | 483.97 | 50.0 | 1035.02 | 53.3 | 1570.14 |
| | Our | 60.0 | **1.13** | 30.0 | **2.66** | 20.0 | **4.34** |
| | Our ($b = 5$) | **100.0** | 1.76 | **100.0** | 5.26 | **100.0** | 9.03 |
| Boltzmann | Star sampling | 80.0 | 71.53 | 30.0 | 148.30 | 26.7 | 226.89 |
| | MinCPD | 20.0 | 185.34 | 10.0 | 390.17 | 6.7 | 612.90 |
| | Our | **100.0** | **0.87** | 60.0 | **1.64** | 46.7 | **2.80** |
| | Our ($b = 5$) | **100.0** | 1.79 | **100.0** | 3.58 | **73.3** | 6.06 |

## 6 CONCLUSION

In this work, we focus on developing efficient algorithms for the task of locating the largest/smallest elements in a factorized tensor. We first propose a general continuous optimization model, which allows for the development of algorithms from an optimization perspective. Then we introduce the MBI approach and combine it with an alternating iterative method to solve the optimization problem, along with a convergence theory. Additionally, a novel block-search strategy is developed to further enhance accuracy. Numerical experiments with synthetic and real-world tensors demonstrate that our proposed algorithm can achieve significant improvements in both accuracy and efficiency over the existing works. Due to the generality of the proposed continuous optimization model, the proposed algorithms could also be applied to other tensor formats, such as Tucker, tensor-train, and tensor ring. Furthermore, there are challenges in estimating the convergence speed of Algorithm 1 and establishing the convergence theory of Algorithm 2. These will be verified and explored as our future works.

---

[2]Retrieved from `https://github.com/tensorly/tensorly/blob/main/tensorly/datasets/data/COVID19_data.npy`

[3]Retrieved from `http://yann.lecun.com/exdb/mnist/`

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

## A  APPENDIX

### A.1  PROOF OF THEOREM 1

*Proof.* Due to the rank-one structure of the optimal solution of problem (2.3), which is equivalent to

$$\max_{\boldsymbol{\mathcal{X}}} \langle \boldsymbol{\mathcal{X}}, \boldsymbol{\mathcal{A}} * \boldsymbol{\mathcal{X}} \rangle$$
$$\text{s.t. } \boldsymbol{\mathcal{X}} = \boldsymbol{x}^{(1)} \circ \boldsymbol{x}^{(2)} \cdots \circ \boldsymbol{x}^{(N)}, \tag{A.1}$$
$$\boldsymbol{x}^{(n)} = 1 \text{ for all } n = 1, 2, \cdots, N.$$

According to the CP representation of $\boldsymbol{\mathcal{A}}$, i.e., $\sum_{r=1}^{R} \boldsymbol{U}_{:,r}^{(1)} \circ \boldsymbol{U}_{:,r}^{(2)} \cdots \circ \boldsymbol{U}_{:,r}^{(N)}$, and the rank-one form of $\boldsymbol{\mathcal{X}}$, then $\boldsymbol{\mathcal{A}} * \boldsymbol{\mathcal{X}}$ can be represented by

$$\sum_{r=1}^{R} (\boldsymbol{U}_{:,r}^{(1)} * \boldsymbol{x}^{(1)}) \circ (\boldsymbol{U}_{:,r}^{(2)} * \boldsymbol{x}^{(2)}) \cdots \circ (\boldsymbol{U}_{:,r}^{(N)} * \boldsymbol{x}^{(N)}).$$

Further, the objective function of problem A.1 can be rewritten as

$$\sum_{r=1}^{R} (\boldsymbol{x}^{(1)T}(\boldsymbol{U}_{:,r}^{(1)} * \boldsymbol{x}^{(1)}))(\boldsymbol{x}^{(2)T}(\boldsymbol{U}_{:,r}^{(2)} * \boldsymbol{x}^{(2)})) \cdots (\boldsymbol{x}^{(N)T}(\boldsymbol{U}_{:,r}^{(N)} * \boldsymbol{x}^{(N)})),$$

which illustrates that problem A.1 is equivalent to (2.3). $\square$

### A.2  PROOF OF THEOREM 2

*Proof.* For convenience, we denote the objective function of problem (2.3) as $f(\boldsymbol{x}^{(1)}, \boldsymbol{x}^{(2)}, \cdots, \boldsymbol{x}^{(N)})$, then $\lambda = f(\boldsymbol{x}^{(1)}, \boldsymbol{x}^{(2)}, \cdots, \boldsymbol{x}^{(N)})$. Let $\mathcal{S}^{(n)}$ be the unit sphere in $\mathbb{R}^{I_n}$ for all $n = 1, 2, \cdots, N$, and $\mathcal{S} = \mathcal{S}^{(1)} \times \mathcal{S}^{(2)} \cdots \times \mathcal{S}^{(N)}$. Since $\boldsymbol{x}^{(n)} \in \mathcal{S}^{(n)}$, the sequence $\{(\boldsymbol{x}_k^{(1)}, \boldsymbol{x}_k^{(2)}, \cdots, \boldsymbol{x}_k^{(N)})\}$ is bounded, and there exists a convergent subsequence $\{(\boldsymbol{x}_{k_l}^{(1)}, \boldsymbol{x}_{k_l}^{(2)}, \cdots, \boldsymbol{x}_{k_l}^{(N)})\}$ such that

$$\lim_{l \to \infty} (\boldsymbol{x}_{k_l}^{(1)}, \boldsymbol{x}_{k_l}^{(2)}, \cdots, \boldsymbol{x}_{k_l}^{(N)}) = (\boldsymbol{x}_*^{(1)}, \boldsymbol{x}_*^{(2)}, \cdots, \boldsymbol{x}_*^{(N)}).$$

Then we have

$$\lim_{l\to\infty} f(\boldsymbol{x}_{k_l}^{(1)},\boldsymbol{x}_{k_l}^{(2)},\cdots,\boldsymbol{x}_{k_l}^{(N)}) = f(\boldsymbol{x}_*^{(1)},\boldsymbol{x}_*^{(2)},\cdots,\boldsymbol{x}_*^{(N)}) = \lim_{l\to\infty}\lambda_{k_l} = \lambda_*.$$

Further, since the sequence $\{\lambda_k\}$ is convergent, we have

$$\lim_{k\to\infty}\lambda_k = \lim_{l\to\infty}\lambda_{k_l} = \lambda_*,$$

which implies that $\{\lambda_k\}$ converges to the $\lambda_*$ that corresponds to the value of $f$ at the limit point $(\boldsymbol{x}_*^{(1)},\boldsymbol{x}_*^{(2)},\cdots,\boldsymbol{x}_*^{(N)})$. According to the strategy of MBI, we know that the following inequality

$$f(\boldsymbol{x}_{k_l}^{(1)},\cdots,\boldsymbol{x}_{k_l}^{(n-1)},\boldsymbol{x},\boldsymbol{x}_{k_l}^{(n+1)},\cdots,\boldsymbol{x}_{k_l}^{(N)}) \le f(\boldsymbol{x}_{k_l}^{(1)},\cdots,\boldsymbol{x}_{k_l}^{(n-1)},\tilde{\boldsymbol{x}},\boldsymbol{x}_{k_l}^{(n+1)},\cdots,\boldsymbol{x}_{k_l}^{(N)}) \le$$
$$f(\boldsymbol{x}_{k_l+1}^{(1)},\cdots,\boldsymbol{x}_{k_l+1}^{(n-1)},\boldsymbol{x}_{k_l+1}^{(n)},\boldsymbol{x}_{k_l+1}^{(n+1)},\cdots,\boldsymbol{x}_{k_l+1}^{(N)}) \le f(\boldsymbol{x}_{k_l+1}^{(1)},\cdots,\boldsymbol{x}_{k_l+1}^{(n-1)},\boldsymbol{x}_{k_l+1}^{(n)},\boldsymbol{x}_{k_l+1}^{(n+1)},\cdots,\boldsymbol{x}_{k_l+1}^{(N)})$$
$$\tag{A.2}$$

holds for all $n = 1, 2, \cdots, N$ and $\boldsymbol{x} \in \mathcal{S}^{(n)}$, where $\tilde{\boldsymbol{x}}$ is the updated value of $\boldsymbol{x}^{(n)}$ when other variables are fixed. Take the limit on both sides of A.2, then for any $\boldsymbol{x}$ in $\mathcal{S}^{(n)}$, we can obtain

$$f(\boldsymbol{x}_*^{(1)},\cdots,\boldsymbol{x}_*^{(n-1)},\boldsymbol{x},\boldsymbol{x}_*^{(n+1)},\cdots,\boldsymbol{x}_*^{(N)}) \le f(\boldsymbol{x}_*^{(1)},\cdots,\boldsymbol{x}_*^{(n-1)},\boldsymbol{x}_*^{(n)},\boldsymbol{x}_*^{(n+1)},\boldsymbol{x}_*^{(N)}),$$

which means that $f$ reaches its maximum value at the point $(\boldsymbol{x}_*^{(1)},\boldsymbol{x}_*^{(2)},\cdots,\boldsymbol{x}_*^{(N)})$ along the $\boldsymbol{x}^{(n)}$ coordinate for all $n = 1, 2, \cdots, N$. Clearly, the gradient of $f$ on the manifold $\mathcal{S}$, i.e., $\nabla_{\mathcal{S}}f$, equals to 0, that is, $(\boldsymbol{x}_*^{(1)},\boldsymbol{x}_*^{(2)},\cdots,\boldsymbol{x}_*^{(N)})$ is the stationary point of problem (2.3). $\square$

### A.3 PROOF OF THEOREM 3

*Proof.* We first rewrite problem (2.3) as

$$\max_{\boldsymbol{y}^{(1)},\cdots,\boldsymbol{y}^{(N)}} g(\boldsymbol{y}^{(1)},\cdots,\boldsymbol{y}^{(N)})$$
$$\text{s.t. } \boldsymbol{y}^{(n)} \in \mathcal{Y}^{(n)} \text{ for all } n = 1,\cdots,N, \tag{A.3}$$

where $g(\boldsymbol{y}^{(1)},\cdots,\boldsymbol{y}^{(N)}) = f(\frac{\boldsymbol{x}_*^{(1)}+\boldsymbol{y}^{(1)}}{\|\boldsymbol{x}_*^{(1)}+\boldsymbol{y}^{(1)}\|_2},\cdots,\frac{\boldsymbol{x}_*^{(N)}+\boldsymbol{y}^{(N)}}{\|\boldsymbol{x}_*^{(N)}+\boldsymbol{y}^{(N)}\|_2})$, and $\mathcal{Y}^{(n)}$ represents the set in $\mathbb{R}^{I_n}$ that is orthogonal to $\boldsymbol{x}_*^{(n)}$. Clearly, the optimal solution of A.3 is $\boldsymbol{0}$. From the convergence theory of MBI illustrated in Theorem 3.3 of Ref. [3], we know that Algorithm 1 is R-linearly convergent to $(\boldsymbol{x}_*^{(1)},\boldsymbol{x}_*^{(2)},\cdots,\boldsymbol{x}_*^{(N)})$ when the Hessian matrix $\nabla^2 g(\boldsymbol{0})$ is negative definite on $\mathcal{Y}^{(1)}\times\mathcal{Y}^{(2)}\cdots\times\mathcal{Y}^{(N)}$. Therefore, we only need to prove that for any $\boldsymbol{y} \in \mathcal{Y}^{(1)} \times \mathcal{Y}^{(2)} \cdots \times \mathcal{Y}^{(N)}$, $\nabla^2 g(\boldsymbol{0})$ satisfies $\langle \boldsymbol{y}, \nabla^2 g(\boldsymbol{0})\boldsymbol{y}\rangle < 0$. By the chain rule, we obtain

$$\langle \boldsymbol{y}, \nabla^2 g(\boldsymbol{0})\boldsymbol{y}\rangle = \langle \boldsymbol{y}, \nabla^2 f_*\boldsymbol{y}\rangle - \sum_{n=1}^{N}\boldsymbol{x}^{(n)T}\nabla f_{*n}\|\boldsymbol{y}^{(n)}\|_2^2, \tag{A.4}$$

where $\nabla f_{*n}$ is the gradient of $f$ at $(\boldsymbol{x}_*^{(1)},\boldsymbol{x}_*^{(2)},\cdots,\boldsymbol{x}_*^{(N)})$ along the $\boldsymbol{x}^{(n)}$ coordinate, which is represented by $\sum_{r=1}^{R}\alpha_r^*(\boldsymbol{U}_{:,r}^{(n)}*\boldsymbol{x}_*^{(n)})$, and $\alpha_r^* = \prod_{l\neq n}(\boldsymbol{x}_*^{(l)T}(\boldsymbol{U}_{:,r}^{(l)}*\boldsymbol{x}_*^{(l)}))$. By the expression of $\nabla f_*$, we have

$$\nabla^2 f_{*n,n} = \sum_{r=1}^{R}\alpha_r^*\text{diag}(\boldsymbol{U}_{:,r}^{(n)})$$

and

$$\nabla^2 f_{*m,n} = \sum_{r=1}^{R}\beta_r^*(\boldsymbol{U}_{:,r}^{(m)}*\boldsymbol{x}_*^{(m)})\circ(\boldsymbol{U}_{:,r}^{(n)}*\boldsymbol{x}_*^{(n)}),\ m\neq n,$$

where $\alpha_r^* = \prod_{l\neq n}(\boldsymbol{x}_*^{(l)T}(\boldsymbol{U}_{:,r}^{(l)}*\boldsymbol{x}_*^{(l)}))$ and $\beta_r^* = \prod_{l\neq m,n}(\boldsymbol{x}_*^{(l)T}(\boldsymbol{U}_{:,r}^{(l)}*\boldsymbol{x}_*^{(l)}))$. Since $(\boldsymbol{x}_*^{(1)},\boldsymbol{x}_*^{(2)},\cdots,\boldsymbol{x}_*^{(N)})$ satisfies the KKT condition corresponding to the problem (2.3), i.e.,

$$\nabla f_{*n} = \sum_{r=1}^{R}\alpha_r^*(\boldsymbol{U}_{:,r}^{(n)}*\boldsymbol{x}_*^{(n)}) = \lambda_*\boldsymbol{x}^{(n)},\ n = 1, 2, \cdots, N,$$

we have

$$\sum_{n=1}^{N} \boldsymbol{x}_*^{(n)T} \nabla f_{*n} \|\boldsymbol{y}^{(n)}\|_2^2 = \lambda_* \sum_{n=1}^{N} \|\boldsymbol{y}^{(n)}\|_2^2. \tag{A.5}$$

Furthermore, we know that the optimal solution $\boldsymbol{x}_*^{(n)}$ is a column of the $I_n \times I_n$ identity matrix, which is denoted as $\boldsymbol{e}_{i_n}^{(n)}$. Then for $m \neq n$, $\nabla^2 f_{*m,n}$ can be reformulated by $\sum_{r=1}^{R} \beta_r^* (\boldsymbol{U}_{i_m,r}^{(m)} \cdot \boldsymbol{U}_{i_n,r}^{(n)}) \boldsymbol{E}^{(m,n)}$, where $\boldsymbol{E}^{(m,n)} \in \mathbb{R}^{I_m \times I_n}$ and satisfies

$$\boldsymbol{E}_{i,j} = \begin{cases} 1, & (i,j) = (i_m, i_n), \\ 0, & \text{otherwise.} \end{cases}$$

And we have

$$\boldsymbol{y}^{(m)T} \nabla^2 f_{*m,n} \boldsymbol{y}^{(n)} = \sum_{r=1}^{R} \beta_r^* (\boldsymbol{U}_{i_m,r}^{(m)} \cdot \boldsymbol{U}_{i_n,r}^{(n)})(\boldsymbol{y}_{i_m}^{(m)} \cdot \boldsymbol{y}_{i_n}^{(n)}).$$

Since $\boldsymbol{y}^{(n)} \in \mathcal{Y}^{(n)}$, which is orthogonal to $\boldsymbol{x}_*^{(n)}$, i.e., $\boldsymbol{y}_{i_n}^{(n)} = 0$, we obtain that $\boldsymbol{y}^{(m)T} \nabla^2 f_{*m,n} \boldsymbol{y}^{(n)} = 0$ holds for all $\boldsymbol{y}^{(m)} \in \mathcal{Y}^{(m)}, \boldsymbol{y}^{(n)} \in \mathcal{Y}^{(n)}$. Combined with A.4 and A.5, we have

$$\langle \boldsymbol{y}, \nabla^2 g(\boldsymbol{0}) \boldsymbol{y} \rangle = \sum_{n=1}^{N} \boldsymbol{y}^{(n)T} \nabla^2 f_{*n,n} \boldsymbol{y}^{(n)} - \lambda_* \sum_{n=1}^{N} \|\boldsymbol{y}^{(n)}\|_2^2, \ \forall \ \boldsymbol{y} \in \mathcal{Y}^{(1)} \times \mathcal{Y}^{(2)} \cdots \times \mathcal{Y}^{(N)}.$$

Due to the optimality of $(\boldsymbol{x}_*^{(1)}, \boldsymbol{x}_*^{(2)}, \cdots, \boldsymbol{x}_*^{(N)})$, the largest eigenvalue of $\nabla^2 f_{*n,n}$ is $\lambda_*$, which implies that

$$\sum_{n=1}^{N} \boldsymbol{y}^{(n)T} \nabla^2 f_{*n,n} \boldsymbol{y}^{(n)} < \lambda_* \sum_{n=1}^{N} \|\boldsymbol{y}^{(n)}\|_2^2$$

for all $\boldsymbol{y} \in \mathcal{Y}^{(1)} \times \mathcal{Y}^{(2)} \cdots \times \mathcal{Y}^{(N)}$, that is, $\nabla^2 g(\boldsymbol{0})$ is negative positive on $\mathcal{Y}^{(1)} \times \mathcal{Y}^{(2)} \cdots \times \mathcal{Y}^{(N)}$. $\quad\square$

