# OpenReview forum: "A-Loc: Efficient Alternating Iterative Methods for Locating the $k$ Largest/Smallest Elements in a Factorized Tensor"
_ICLR.cc/2024/Conference — Submitted to ICLR 2024_

### Official Review · Reviewer_R5w8 · 2023-10-30

**Soundness:** 3 good
**Presentation:** 3 good
**Contribution:** 1 poor
**Rating:** 1
**Confidence:** 4

**Summary:**

The paper takes several well-known ideas, in particular maximization/minimization over a continuous $x$ in a discrete problem, and what the paper calls block-search strategy (a.k.a. top-k). These approaches are used to simultaneously find the minimum or maximum of a CP-tensor. Several theorems in the paper do not fully shed light on the accuracy of the method, the evaluation of which remains more empirical. Numerical experiments are set on random low-dimensional CP-tensors, CP-factorization of two well-known model fucntions, and CP-factorization of several small real-world datasets.

**Strengths:**

- The paper is generally well structured and written
- In some cases good results are obtained when searching for CP-tensors minimum

**Weaknesses:**

- The paper takes known algorithms, and only connects them together in the special case of the problem of finding the extremum of CP-tensors.

- Comparisons in numerical examples are given with only two (and in the case of finding the minimum, only one) other method, which is clearly insufficient.
   * The coloring in the tables when both the presented method and its competitor are the best is given only for the presented method. It would be fair to color all cases when a particular method was the best. For example, in Table 2, color all cases of minimum for both functions, since they are equal for the presented method and for MInCPD.

- Table 1, which compares methods for random vectors, is not entirely honest. Indeed, the _star sampling_ method can only find maximum (modulo) elements of CP-tensor. But once one is found, let its values be $m$, we can do a transformation $m \mathcal E - \mathcal A$ (very similar to the one described on page 2 in section 2 of the paper), which increases the CP-rank by only 1. Then the problem of finding the minimum in the initial tensor $\mathcal A$ reduces to finding the maximum in this tensor. Unfortunately, I haven't found open source code for the _star sampling_ method, but I suspect it would work (with this trick) just as well as the presented method. And in the problem of maximum finding the _star sampling_ algorithm, as the authors honestly write, already wins, so the the contribution made by this new approach may be considered negligible in this case.

- As for finding optima of the Rastrigin and Schwefel model functions, this and many other functions were used to test a similar algorithm optima_tt, which is cited in this paper as Chertkov et al. (2022). If we take the _optima_tt_ code, which is open source, and apply it to TT-tensors that exactly describe the two functions (with exact TT-ranks of 2 each) with the same parameters (n=4096, d=10), we get the following results for the minimum and for the maximum (I took the hyperparameter $k=5$):
Rastrigin: min: 0.0014192582... max: 403.5327475...
Schwefel min: 0.000725485....  max: 8379.65727....
Thus, for these two functions the combination of using CP-compression and the newly presented method is meaningless.

- No code is provided so that experiments can be repeated

- The Theorems in the paper are rather trivial, one of them is not proved at all (see below). Thus, there is no serious analysis of convergence. To be fair, it should be noted that there is no such analysis for many other similar methods.

- in the proof of Theorem 1, in the last equation in (A.1) it is not clear what "$x^{(n)}=1$" means. If it is a constant vector, how can we search for $\max$ w.r.t. it?
If it $\||x^{(n)}\||_2=1$ then (A.1) is not just equivalent (as the proof of the theorem states), but coincides with problem (2.3).
But Theorem 1 asserts more than that. It asserts that for continuous values of $x^{(n)}$, i.e., $x^{(n)}\in\mathbb R^{I_n}$, the solution of the maximization problem (2.3) coincides with the maximum of CP-Tensor element, i.e., with the case where each vector $x^{(n)}$ are binary and represents (different) rows of the unit matrix. In this form, Theorem 1 has never been proved in the this paper.

- times in the Table 4 with experiments with "real-world" data are not relevant, because it will be much faster, for example, for COVID dataset expand it the full tensor, and then search the maximum. Moreover, the compression time of these real-world datasets into CP format is not taken into account.
    * link the mnist dataset is not valid -- login and password requires to access it.

- at the beginning (page 3) the paper says that the algorithm parameter-free. But then it turns out that the final Algorithm has an adjustable parameter $b$.

Minor:

Suggestion:  put brackets around formula references (use "\eqref" instead of "\ref").

**Questions:**

- Have you tried experiments on model functions other than Rastrigin and Schwefel?
- Have you tried scaling the method to significantly higher dimensions ($N=100,1000$), which overcome the curse of dimensionality?

---

> ### Author Response · Authors · 2023-11-11
> **Reply to Reviewer R5w8**
>
> - One of the contributions of this work is to transform the problem of finding the largest element of a factorized tensor into a continuous optimization problem, i.e., Theorem 1, which makes it possible to design algorithms for finding the largest element of a factorized tensor from a continuous optimization perspective. More importantly, our proposed continuous optimization model is applicable to other formats such as Tucker and tensor-train. For example, if the tensor is given in Tucker format, i.e., $\mathcal{A}=\mathcal{G}\times_{1}U^{(1)}\times_{2}U^{(2)}\cdots\times_{N}U^{(N)}$, then the corresponding optimization problem can be written as
> $$\max\limits_{x^{(1)},x^{(2)},\cdots,x^{(N)}}\mathcal{G}\times_{1}(x^{(1)T}(U^{(1)}\odot x^{(1)}))\times_2(x^{(2)T}(U^{(2)}\odot x^{(2)}))\cdots\times_{N}(x^{(N)T}(U^{(N)}\odot x^{(N)})),\ \text{subject to}\ ||x^{(n)}||_2=1\ \text{for all}\ n=1,2,\cdots,N.$$
> - We have listed and compared the algorithms in the existing works. If there are any omissions, please point them out.
> - In the star sampling method, the top-k elements are obtained from the samples directly, but it may have too few effective samples that only include the largest element. Therefore, for a fair comparison, we use the same way to find the top-k elements in our experiments. Although the star sampling method is more accurate for random tensors, it requires many samples and is less efficient. Additionally, experiments in Sec. 5.2 and 5.3 show that the star sampling method is not always accurate.
> - The CP formats of Rastrigin and Schwefel can be directly obtained by Eq. (5.1), and their CP-ranks are both $d+1$. Since CP and TT formats can be converted to each other, they can also be expressed in TT format. As mentioned before, our proposed algorithms are also applicable to the tensor given in TT format due to the generality of the proposed continuous optimization model.
> - The key to Theorem 1 is that we observe the tensor corresponding to the solution is rank one, thus the problem (2.2) can be further rewritten as (2.3).
> - The goal of experiments in Sec. 5.3 is to verify the feasibility of our proposed algorithms, thus we use the data that can be stored in a laptop. However, for the higher-order tensor in practical applications, it can not be explicitly stored, and our proposed algorithms still work due to the fact that the time and memory cost of each iteration of them are $\mathcal{O}(RI_n)$, which is linear with the order $N$.

---

> ### Comment · Reviewer_R5w8 · 2023-11-20
>
> I thank the authors for their rather detailed answers. I will answer point by point (each of my black bullet points corresponds to the authors' bullet points)
>
>
> *
> 	- No, the problem of finding the extremal element for tensors in CPD using continuous elements is proposed in the paper Nicholas D Sidiropoulos, Paris Karakasis, and Aritra Konar. _Finding the smallest or largest element of a tensor from its low-rank factors._ arXiv preprint arXiv:2210.11413, you cited in your work.
> Compare formula (1) from this paper, where the positive values of $p$ coincide with the squares of $x$ from (2.3) of the formula of your paper.
> Without considering the notation, the formulas are equivalent.
>
>  * Moreover, you did not prove your theorem, because a) it is not clear (you did not answer my question in the review) what $x^{(n)}=1$ means b) you did not consider the case when a tensor has two or more extremal elements. Compare it with the proof of a similar theorem from the paper Sidiropoulos _et. al._ , it is already complete there. In Appendix you just proved the equivalence of (A.1) and (2.3), and both of these formulas contain continuous variables, so there is no reduction to the initial discrete task.
>
>  - The method from the Sidiropoulos _et. al._  paper extends naturally to Tucker: $\max_\limits{ [p_n\geq0, \hbox{ }\textbf{1}^T p_n=1]_{n=1}^N   } \mathcal G \times_1 (p_1^T U^{(1)})\times_2 (p_2^T U^{(2)})\cdots\times_N (p_N^T U^{(N)})$ so you're not the first here either.
>  - To summarize, you haven't invented a new method, you've assembled yours as a kit from parts of existing ones.
>
>
> * The main omission is the lack of real-world problems.  If you are talking about in principle low-rank data, as in the functions example, then you need to compare to other low-rank formats, such as TT decomposition as in the paper you cited by Andrei Chertkov, _et. al._ (arXiv:2209.14808), where maximum TT decomposition is used to find the extremum in discretized Rastrigin, Schwefel and other 8 more complex functions.  But most importantly, if one presents a maximum search method in CPD format, then it is author's job to find real examples where the new method would show superiority. I.e. it is important not just to show how the method works on extremely artificial examples, but to show its superiority in a problem where using CPD (and __not__ Tucker, TT and others) would be justified. Otherwise, it is much easier for one to use other decompositions for which similar methods of extremum search have already been developed.
>
> * Since the method presented in the paper is a combination of several heuristics, it is not very fair to compare it with one "pure" method. As an addition to Table 1 it would be interesting to see a comparison with a rather natural additional heuristic to Star sampling that converts the minimum of a CD tensor into a maximum. Moreover, a very similar transformation is given in this paper.
>
> * _Any_ tensor can be converted to _any_ format with _any_ given accuracy, the question is in ranks. For this particular case, discretization of Rastrigin and Schwefel functions, the ranks of exact TT-decomposion is 2, which is smaller, then $(d+1)$. If you know an efficient method to directly convert a CPD decomposition with ranks $(d+1)$ to a TT decomposition with rank 2 without loss of accuracy, please let me know. But it is not so important, what is important is that in the case of these functions, the TT-decomposition has $<4nd$ parameters, while the CPD has $(d+1)nd$ , which for the setup in your paper $d=10$, $n=4096$ gives $163840$ and $450560$ respectively. Thus, using CPD for this case of these functions is both inefficient and uneconomical, and, as I showed in the main review, gives worse accuracy than the existing algorithm for finding extrema in TT-tensors. Thus, the practical significance and novelty are not substantiated by these examples.
>
> * As I wrote above, this particular result was already in the Sidiropoulos _el. al._ paper.
>
> * All tensor decompositions (CPD, Tucker, TT, HT, etc.) are designed to break the curse of dimensionality. My concern was about the practical relevance of the work. Compressing a fairly small array into a CPD (which already takes a certain amount of time and loses accuracy) and then looking for an extremum in that decomposition is of no practical significance. Especially since there was not even a comparison in terms of speed (since you place the running time in the table) with direct search for minimum or maximum using numpy.min and numpy.max. If your paper would be focused on theoretical calculations, or would contain a fundamentally new algorithm, then it would be possible to pay less attention to the practical section. But your algorithm is just a set of already known tricks that is aimed at practical use, and in such a setting real-world problems are needed, not quite simple and extremely artificial synthetic ones.
>
>
> Finally, I keep my negative rating and increase confidence in it.

---

### Official Review · Reviewer_TDnR · 2023-10-31

**Soundness:** 2 fair
**Presentation:** 3 good
**Contribution:** 2 fair
**Rating:** 5
**Confidence:** 3

**Summary:**

This paper presents a algorithm for locating the largest and smallest elements in a tensor with low-rank format. The algorithm combines an alternating iterative method and a maximum block increasing approach, along with a newly-designed block-search strategy. The proposed algorithm achieves improvements in accuracy and efficiency compared to existing methods, as demonstrated through numerical experiments. It also discusses the use of the algorithm in various low-rank representations and highlights its advantage of obtaining both the largest/smallest elements and their location simultaneously.

**Strengths:**

*1. Quality:*  The paper presents an algorithm that significantly improves upon existing methods by achieving remarkable advances in both accuracy and efficiency. The comparisons with existing methods underscore the superior performance of the proposed approach in terms of both accuracy and computational efficiency. Moreover, the paper discusses the algorithm's applicability to various low-rank representations, including CP, Tucker, and TT formats.

*2. Clarity:*  The authors employ precise mathematical notations consistently, facilitating the understanding of the proposed algorithm, theoretical analysis, and numerical experiments.

**Weaknesses:**

1. *Limited Novelty:* The paper draws inspiration from prior work and utilizes established techniques. It doesn't clearly differentiate how the proposed algorithms significantly advance beyond existing methods, both conceptually and theoretically. A more thorough exploration of the novelty would strengthen the paper's contribution.

2. *Limited Impact:* The paper's contribution is niche, addressing a specific task related to low-rank tensors. It lacks the potential to significantly impact or disrupt the broader machine learning community. A broader discussion of potential applications or extensions could enhance its relevance.

3. *Lack of Code Availability:* The absence of publicly available code for replicating the proposed algorithm limits reproducibility and practical usability for the research community. Providing code would improve transparency and facilitate further research.

4. *Inadequate Discussion of Limitations:* The paper does not thoroughly discuss potential limitations or challenges associated with applying the proposed algorithm in practical scenarios. A comprehensive examination of limitations is essential for practical decision-making.

**Questions:**

*Question 1:* Can the authors provide a more detailed explanation of how your proposed algorithms significantly differ from existing methods in both conceptual and theoretical aspects? Are there specific aspects of your approach that are novel and distinct from prior work that you draw inspiration from?

*Question 2:* Could the authors elaborate on potential applications or extensions of your method beyond the specific task of low-rank tensors? How might it contribute to broader machine learning challenges? What are the limitations of your method in terms of scalability or adaptability to different problem domains?

---

### Official Review · Reviewer_nzvV · 2023-10-31

**Soundness:** 1 poor
**Presentation:** 1 poor
**Contribution:** 1 poor
**Rating:** 1
**Confidence:** 4

**Summary:**

The paper presents an approach for computing the largest elements of a factorized tensor by means of alternating optimization.

**Strengths:**

+ The results regarding convergence of the method would make the approach valuable for a reasonably formulated problem.

**Weaknesses:**

- The paper seems to obfuscate at least two major technical oversights:
    - The symmetric tensor eigenvalue formulation of the problem is used, by different factor matrices are optimized by alternating optimization. In the symmetric tensor eigenvalue problem, each component of the rank-1 factorization is typically set to be the same. Symmetry is mentioned in the problem definition in Section II, but never thereafter.
    - The proof of theorem 1 does not appear complete or correct to me. Convergence is considered based only the objective function, which is convergent for any method that has a nondecreasing and bounded objective. Hence, the contrast to convergence of standard alternating least squares does not make sense. See, Uschmajew, A. (2012). Local convergence of the alternating least squares algorithm for canonical tensor approximation. SIAM Journal on Matrix Analysis and Applications, 33(2), 639-652.

I believe the paper should not be published due to technical errors that are obfuscated by the writing.

**Questions:**

None

---

> ### Author Response · Authors · 2023-11-11
> **Reply to Reviewer nzvV**
>
> - We think it necessary to state that we consider the problem of finding the largest/smallest elements of a factorized tensor in this work, which can be equivalently transformed into a symmetric eigenvalue problem, where the symmetric matrix $A\in\mathbb{R}^{n_1n_2\cdots n_d\times n_1 n_2\cdots n_d}$ is a diagonal matrix consisting of the entries of the tensor $\mathcal{A}$. Based on this, we further combine the rank-one structure of the eigenvector to propose a continuous optimization model and use an alternating iterative method to solve it.
>
> - The goal of Theorem 1 is to show the equivalence of finding the largest element of $\mathcal{A}$ and the proposed continuous optimization problem, which can be easily proved due to the rank-one structure of the eigenvector. Our convergence analysis involves two parts. The first one is global convergence, i.e., Theorem 2, which shows that Algorithm 1 converges to a stationary point, which matches the theory established in B. Chen, et al., SIAM Journal on Optimization, 22(1):87–107, 2012. The second one is local convergence, which requires a very careful analysis of the property of the objective function at the stationary point, just like A. Uschmajew analyzes the convergence of alternating least squares algorithm for canonical tensor approximation.

---

### Official Review · Reviewer_neUt · 2023-11-06

**Soundness:** 3 good
**Presentation:** 3 good
**Contribution:** 2 fair
**Rating:** 3
**Confidence:** 4

**Summary:**

In this paper, the authors develop algorithms aimed at identifying the largest and smallest elements in a factorized tensor. They propose an alternating iterative algorithm, which is enhanced with a maximum block increasing approach and a novel block-search strategy. The validation of their method is provided through both theoretical analysis and experimental evaluations.

**Strengths:**

This paper presents a novel approach to a problem arising from tensor analysis - locating the largest/smallest elements in a tensor in the low-rank format. The strengths of the paper can be highlighted as follows:

1. The authors propose a continuous optimization model that is different from the existing methods. This model could be applied to various low-rank representations such as CP, Tucker, and TT formats.

2. The proposed method demonstrates significant improvements in both accuracy and efficiency over existing works in numerical experiments. This suggests that the method may be practically useful in real-world applications.

**Weaknesses:**

While the paper presents a novel approach to locating the largest/smallest elements in a tensor in a low-rank format, some weaknesses mainly related to the theoretical results can be identified:

1. Theorem 1 formulates an equivalent optimization problem for finding the largest element in a tensor, which is a crucial step in the authors' method. However, the paper does not provide an in-depth analysis of this equivalent problem. For example, it would be beneficial to understand the number of minimizers that this problem may have. Such analysis could provide insights into the complexity and potential pitfalls of the optimization problem, and could be critical for the algorithm's performance.

2. Theorem 2 shows that the proposed algorithm's subsequence converges to a stationary point of the optimization problem. However, it's unclear whether the entire sequence converges, and more importantly, whether it converges to the optimal solution. The absence of these guarantees may limit the algorithm's reliability and effectiveness in practice.

3. While Theorem 3 establishes a linear convergence rate to the optimal solution, it does so under the assumption that the initial point is sufficiently close to the optimal solution. Regrettably, the paper falls short in providing a discussion on how to obtain such an initial point. Furthermore, the paper does not explore the potential existence of multiple optimal solutions, a factor that could significantly affect the algorithm's convergence behavior.

**Questions:**

Please answer the questions in Weakness part.

---

> ### Author Response · Authors · 2023-11-11
> **Reply to Reviewer neUt**
>
> - Theorem 1 is one of the main contributions of this work, which makes it possible to design algorithms for finding the largest element of a factorized tensor from a continuous optimization perspective. Analysis of the number of minimizers is equivalent to the analysis of the convergence speed, which is very challenging for the alternating iterative algorithm in the field of optimization. Even for the alternating iterative algorithm for linear problems, i.e., Gauss-Seidel, the estimation of its convergence rate is also very difficult.
>
> - Since the corresponding objective function sequence is monotonic, if there is a subsequence converges to a stationary point of the optimization problem, the objective function sequence converges to the value corresponding to the stationary point. Theorem 2 is a global convergence theory, we cannot guarantee that the alternating iterative method converges to the global optimum, which matches the theory established in B. Chen, et al., SIAM Journal on Optimization, 22(1):87–107, 2012. If the initial point is sufficiently close to the optimal solution, then the stationary point that the alternating iterative method converges to is the optimal point, see Theorem 3.

---

> > ### Comment · Reviewer_nzvV · 2023-11-21
> >
> > Theorem 1 is a trivial application of tensor algebra / CP format.
> >
> > The proof of Theorem 2 assumes a sequence of points exist that approaches the minimum, then argues in terms that the objective value decreases monotonically and hence will converge. This is true for any alternating optimization approach in the sense that any monotonic sequence has a limit.
> >
> > The proof of Theorem 3, refers to "Ref 3", though references are not numbered. But the above issues with theorem 2 to me are already critical.

---

> > > ### Comment · Reviewer_neUt · 2023-11-22
> > >
> > > I concur with the comments made by Reviewer nzvV. The theoretical results presented in the paper require significant improvement.

---

### Meta-Review · Area_Chair_gkyQ · 2023-12-09

**Metareview:**

This paper attempts to find the largest or $k$ largest elements of a tensor written in the form of an exact CP decomposition. This is transformed into a continuous optimization problem subject to unit norm constraints, although this is not the contribution of this paper. A maximum block improvement algorithm is proposed with some convergence analysis given.

**Justification For Why Not Higher Score:**

The studied problem has limit interest. The convergence analysis is not rigorous enough.

**Justification For Why Not Lower Score:**

N/A

---

### Decision · Program_Chairs · 2024-01-16

Reject